# A hepatitis B virus (HBV) sequence variation graph improves alignment and sample-specific consensus sequence construction

Dylan Duchen [1,2]*, Steven J. Clipman[3], Candelaria Vergara[1], Chloe L. Thio[3], David L. Thomas[3], Priya Duggal[1], Genevieve L. Wojcik[1]

**1** Department of Epidemiology, Johns Hopkins Bloomberg School of Public Health, Baltimore, MD, United States of America, **2** Center for Biomedical Data Science, Yale School of Medicine, New Haven, CT, United States of America, **3** Division of Infectious Diseases, Johns Hopkins University School of Medicine, Baltimore, MD, United States of America

* Dylan.duchen@yale.edu

**Data Availability Statement:** The longitudinal HBV sequencing data utilized in this study is available as an NCBI BioProject under the accession

## Abstract

Nearly 300 million individuals live with chronic hepatitis B virus (HBV) infection (CHB), for which no curative therapy is available. As viral diversity is associated with pathogenesis and immunological control of infection, improved methods to characterize this diversity could aid drug development efforts. Conventionally, viral sequencing data are mapped/aligned to a reference genome, and only the aligned sequences are retained for analysis. Thus, reference selection is critical, yet selecting the most representative reference *a priori* remains difficult. We investigate an alternative pangenome approach which can combine multiple reference sequences into a graph which can be used during alignment. Using simulated short-read sequencing data generated from publicly available HBV genomes and real sequencing data from an individual living with CHB, we demonstrate alignment to a phylogenetically representative 'genome graph' can improve alignment, avoid issues of reference ambiguity, and facilitate the construction of sample-specific consensus sequences more genetically similar to the individual's infection. Graph-based methods can, therefore, improve efforts to characterize the genetics of viral pathogens, including HBV, and have broader implications in host-pathogen research.

## Introduction

Approximately one-third of the world's population has been exposed to the hepatitis B virus (HBV), a major cause of hepatocellular carcinoma and end-stage liver disease [1]. With nearly 300 million individuals suffering from chronic HBV infection (CHB), novel drugs are needed as no effective curative therapies currently exist [2]. While spontaneous recovery occurs, the biological mechanisms underlying the immunological control of HBV remain unclear. In addition to age, clinical and environmental factors, and host genetic variation [3, 4], viral genetic diversity contributes to the pathogenesis and the severity of CHB [5–8].

PRJNA479693. Simulated HBV sequencing data and the HBV reference graph have been deposited on Zenodo and can be accessed using the following doi: 10.5281/zenodo.10018711.

**Funding:** This work would not have been possible without the financial support of the following entities: National Institutes of Health (NIH) grant R01AI148049 (D.L.T., P.D., G.L.W) National Institutes of Health (NIH) grant DP2DA056130 (S. J.C.) National Institutes of Health (NIH) COVID-19 supplement grant R01AI148049 (D.L.T., P.D., G.L. W.) Burroughs-Wellcome Fund, MD-GEM training grant (D.D.). National Human Genome Research Institute (NHGRI) grant R35HG011944 (G.L.W.). The funders listed above played no role in the study design, data collection and analysis, decision to publish, or preparation of the manuscript.

**Competing interests:** The authors have declared that no competing interests exist.

HBV has a small (3.2 kilobases (kb)) partially double-stranded circular DNA genome with four overlapping gene-encoding regions and a higher mutation rate than most double-stranded DNA viruses [9]. With 10 known genetically and geographically distinct HBV genotypes and >30 subgenotypes, CHB is also caused by recombinants or mixtures of genotypes [10–16]. Additionally, like other chronic viral infections, intra-host CHB diversity involves multiple viral strains that evolve, mutate, and change in frequency over time, termed a viral quasispecies [17–20]. Intra-host diversity has also been shown to influence disease progression [21–23], treatment outcome [24, 25], and confound molecular epidemiology or surveillance efforts (e.g., transmission of non-consensus/minor-frequency viral strains) [26–28].

Characterizing this extensive genetic variation is therefore important for advancing our understanding of the natural history of disease and potential treatment targets. Sequencing-based analyses of HBV and other microbial pathogens usually involve an initial alignment of sequencing data to a representative reference genome to infer a 'consensus' sequence which provides an approximation of the genome causing the infection. The construction of consensus sequences is a typical objective of viral-focused genetic analyses [29, 30], including HBV [31, 32]. Choosing the right reference is critical, as only data sufficiently similar to the reference can be aligned and retained within subsequent analyses [33, 34]. However, it can be difficult to select the most appropriate reference sequence when analyzing clinical CHB samples of unknown HBV genotype or subgenotype.

The use of unrepresentative reference sequences can interfere with characterizing pathogen diversity, resulting in false or missing mutations and biased phylogenetic relationships [35–37]. Reference selection can also affect the ability to accurately derive sample-specific 'consensus' sequences which reflects the most commonly observed nucleotide at each site across the genome, inferred from the aligning sequencing data against a specific reference sequence. This issue of reference ambiguity is especially problematic for CHB, as a set of phylogenetically representative HBV reference sequences has only recently been proposed [38]. Furthermore, given the extreme diversity of CHB, the use of a single reference sequence, even of the correct HBV genotype, may be insufficient [34, 35].

Approaches involving the simultaneous use of multiple references during read alignment could avoid these issues. One such approach involves combining a phylogenetically representative set of genome-length HBV sequences into a single pangenomic 'genome graph', and using this for alignment rather than any single linear HBV genome sequence. By genome graph, we mean a sequence variation graph comprised of 'nodes', which reflect stretches of genetic sequence, and 'edges', which form the connections between nodes and determine the order a genome-length sequence traverses the graph [33, 36, 39, 40]. Sequence variation graphs efficiently represent genetic variation from multiple genomes and have been shown to improve sequence alignment and variant calling for highly variable regions of the human genome and microbial organisms like *Escherichia coli*. [33, 36, 41]. A graph-based reference containing a representative sampling of the genetic variation observed across all known HBV genotypes/subgenotypes might improve sequence alignment and variant calling, as well as enable the generation of accurate sample-specific consensus sequences for HBV-related genetic analyses. However, whether a graph-based approach can improve viral sequence alignment and sample-specific consensus sequence construction has, to our knowledge, yet to be demonstrated.

Assessing every potential HBV reference sequence to identify the most appropriate reference for a given sample is both computationally expensive and can still fail within the context of recombinant or mixed infections. In this study, we leverage 2,837 publicly available full-length HBV genomes, simulated high-throughput sequencing data from these HBV sequences, and real-world longitudinally collected sequencing data from an individual with CHB to identify the optimal alignment method as determined by the proportion of successfully aligned

HBV sequencing data. To model alignment using standard linear reference sequences, a non-overlapping set of 44 phylogenetically-representative HBV genome sequences reflecting the known breadth of HBV diversity were used [38]. To model graph-based alignment, a HBV reference graph was constructed using these same sequences. By comparing the proportion and accuracy of successfully aligned HBV sequence and alignment-derived consensus sequences, respectively, the accuracy of graph vs. linear reference-based alignment methods will be evaluated to determine whether a phylogenetically diverse graph-based reference can improve genetic analyses of HBV while alleviating the issues associated with reference selection.

The aim of this study is to assess whether a pangenomic graph-based approach can improve sequence alignment and mitigate reference ambiguity for genetic studies of HBV. Additionally, we aim to determine if this approach leads to more accurate sample-specific consensus sequences compared to standard linear reference-based approaches.

## Results

### Simulations to assess HBV sequence-to-graph alignment and coverage

To assess whether the graph or any of the linear reference sequences can capture the known genetic diversity of HBV, short Illumina-like reads were simulated from 59 genetically diverse HBV genomes encompassing 9 HBV genotypes and aligned to both the HBV reference graph and each of the 44 linear references. Despite all reads being of HBV origin (N = 500,002 reads), only 84.3% to 96.6% of sequences successfully aligned to these 44 linear references (**Fig 1A**). In contrast, >99.9% of this diverse simulated HBV sequencing data successfully aligned to an HBV reference graph. To ensure that loci from across all HBV genomes used to create the graph are adequately represented by the reference graph, seven randomly subsampled sets of simulated high-throughput HBV sequencing data (N = 507,938 reads) generated from these 44 genomes were aligned to the HBV reference graph, with 100% of reads always aligning to the graph.

The simulated HBV sequences that failed to align to the linear references (3.4%-15.7%) were not uniformly distributed across the genome, with loci observed to have precipitous drops in coverage corresponding to regions of increased genetic diversity (**Fig 1B**). While these genomic segments were positionally consistent, drops in coverage were highly heterogeneous in magnitude across reference sequences of different HBV genotypes, with the lowest proportion of successfully aligned HBV sequencing data and the most precipitous drops in coverage in regions of increased nucleotide diversity occurring for HBV genotypes H and G reference sequences. As >99.9% of sequencing data successfully aligned to the HBV reference graph, no significant drops in coverage were observed.

To reflect a more realistic scenario in which an individual would likely only have sequences derived from endemic co-circulating genotypes, we limited the simulated data used in our alignment comparisons to those generated from HBV sequences of genotypes B or C, the most common genotypes in East and Southeast Asia [42]. While >99.9% of simulated reads from genotypes B and C successfully aligned to the reference graph, a high proportion of these reads also successfully aligned to linear reference sequences of genotype B (97.9%-98.4%) and C (97.6%-98.8%) (**S3 Fig in S1 File**). This improved performance of the linear references is to be expected given the more homogenous set of simulated HBV reads being assessed.

To determine whether graph-aligned HBV sequences aligned best to the path/reference sequence embedded within the graph of the same HBV genotype as the query sequence, all full-length HBV genome sequences (N = 2,837) and each set of simulated short-read HBV sequencing data generated from the HBV genomes used in the simulations (N = 59) were aligned to the HBV reference graph. For each genotype-specific set of alignments, all genome-

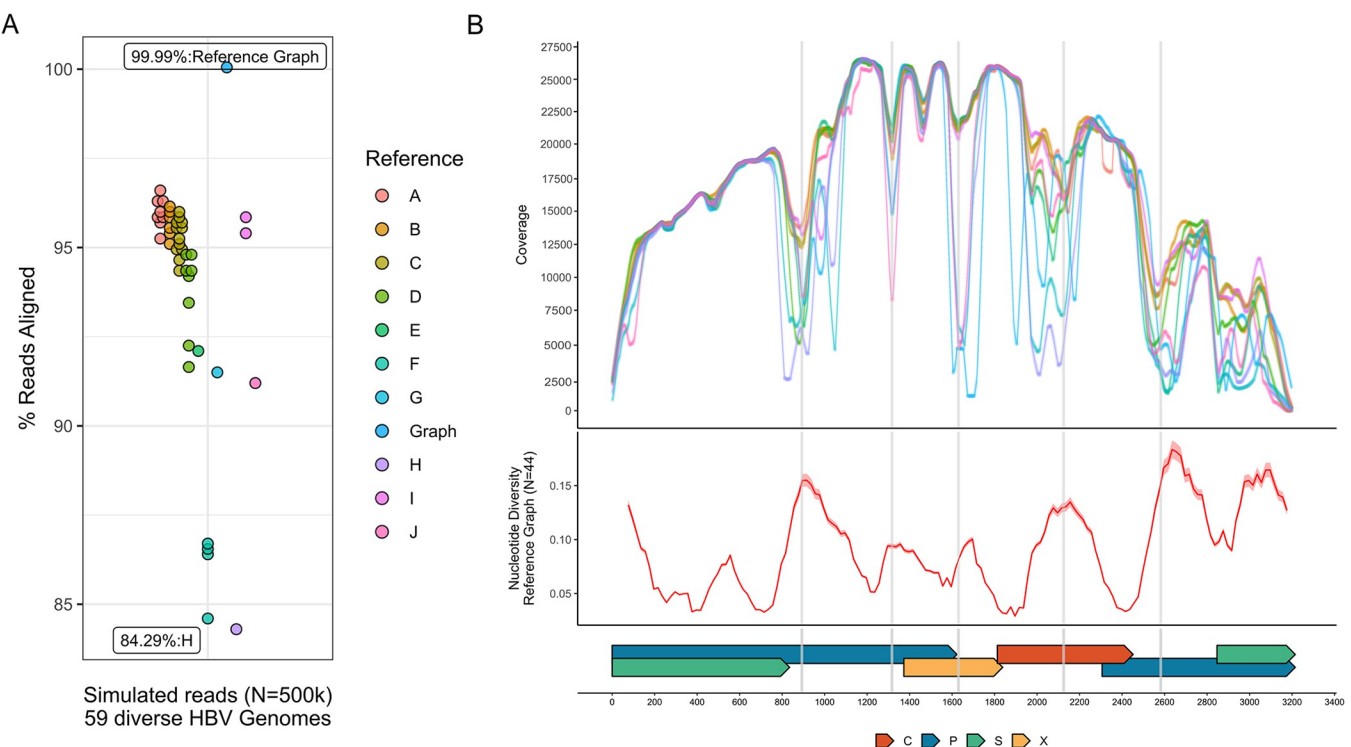

**Fig 1. Alignment and depth of coverage across the HBV genome for simulated HBV sequencing data.** In panel A, points reflect the proportion of successfully aligned simulated reads, colored by either the genotype of the reference or whether graph-based alignment was performed. The Y axis reflects the proportion of successfully aligned reads. Labels indicate the reference sequence genotype or graph used in the alignment and the proportion of reads aligned. In panel B, the X axis reflects genome position, with gene regions provided as colored bars along the base of the figure: C = core, P = polymerase, S = Surface, X = X. In the top section, the Y axis reflects depth of coverage across the HBV genome obtained when using reference sequences of different HBV genotypes, using the same genotype-specific color scheme as panel A. Significant drops in coverage are indicated with the grey vertical lines. The Y axis of the middle section reflects the average pairwise nucleotide diversity (0–1) estimated from all combinations of the included HBV genotypes/subgenotypes using a sliding window across the genome.

length and simulated set of short-read sequences of genotypes A (N = 259, N = 7), B (N = 687, N = 17), C (N = 1,094, N = 19), D (N = 549, N = 9), E (N = 145, N = 2), F (N = 80, N = 2), G (N = 3, N = 1), H (N = 11, N = 1), I (N = 9, N = 1) always resulted in paths of the same HBV genotype having the highest alignment score (**S1 Table in S1 File**), demonstrating the importance of representing each phylogenetically-distinct HBV genotype within the reference graph. These results also demonstrate that the path-specificity of sequence-to-graph alignment can enable HBV genotype prediction using either the alignment score directly or a metric based on the path-depth of nodes with successful alignments for genome-length and high-throughput HBV sequences, respectively.

## Alignment of real CHB sequencing data to an HBV reference graph

Unlike our simulated HBV sequencing datasets, real-world CHB-derived HBV sequencing data can reflect extensive genetic variation due to both host and pathogen-derived evolutionary pressures in addition to any sample processing or sequencing-related errors. Additionally, real CHB sequencing data can have highly variable quality and coverage distributions across the genome. Using the baseline and subsequent longitudinally collected samples from a treatment naïve individual with CHB, analyses of patient-derived CHB sequencing data demonstrates graph-based sequence alignment consistently achieved higher proportions of successfully aligned HBV sequence data compared to any single linear reference (N = 44), with

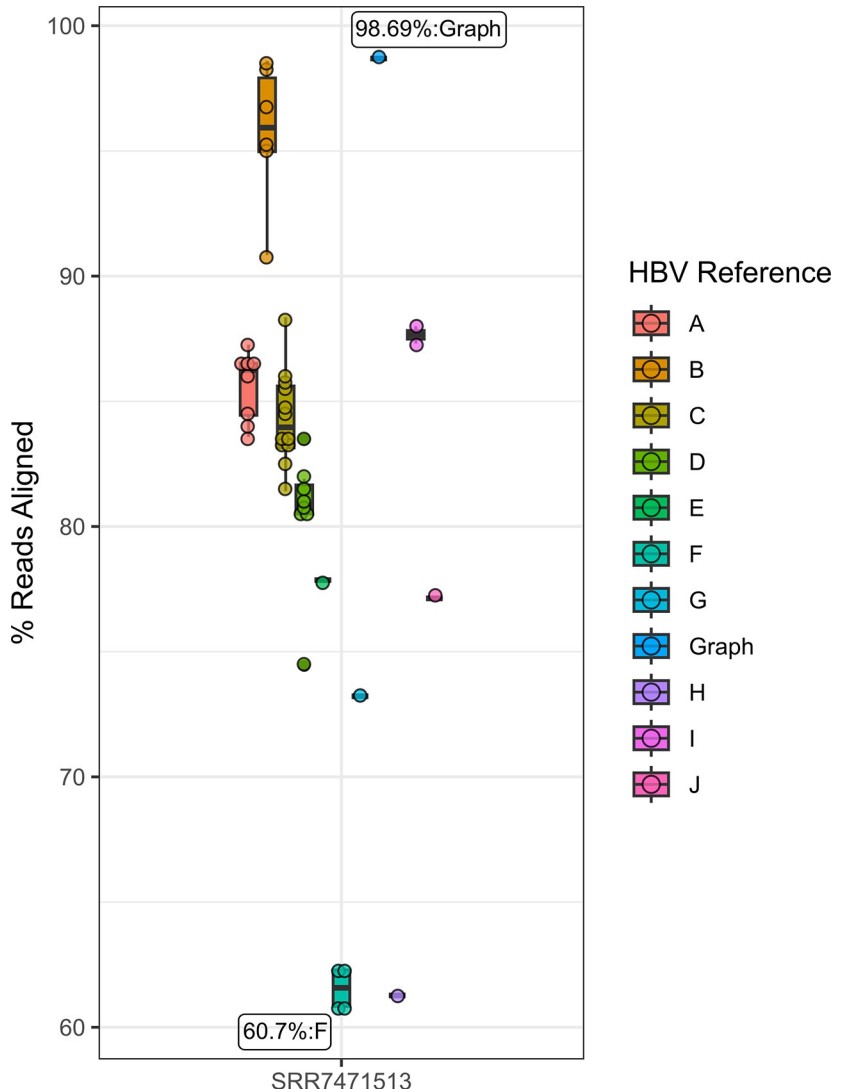

**Fig 2. Proportion of successfully aligned HBV sequencing data for the first CHB sample.** Points reflect the proportion of successfully aligned sequences, colored by either the genotype for linear reference-based alignment or if sequences were aligned to the HBV reference graph. The Y axis reflects the proportion of successfully aligned reads. Labels reflect for the reference of the highest and lowest observed aligned proportions.

98.7%, 93.2%, 99.4%, 98.6%, and 98.7% of successfully aligned sequence for the baseline patient sample (SRR7471513, obtained in 1991) and subsequent samples SRR7471501 (7 months follow-up), SRR7471502 (16 months follow-up), SRR7471499 (30 months follow-up), and SRR7471500 (64 months follow-up) (**Fig 2**), respectively (**S4 Fig in S1 File**). The choice of linear reference had a significant effect on the proportion of aligned sequences across the five samples, with a per-sample difference between the best and worst-performing linear reference ranging from 32.8% to 38.1%. The best performing linear reference (HBV subgenotype B2, GenBank ID: GU815637) resulted in 98.6%, 92.4%, 99.3%, 98.5%, and 98.6% of aligned sequences for each sample (from baseline to final sample, respectively), which were all lower than the proportion of successful graph-based alignments. Notably, differences were also observed between references of the 'correct' HBV genotype (B), with a per-sample difference between the best and worst-performing reference ranging between 7.2% (85.3% vs. 92.4% for

the second sample) and 7.8% (90.8% vs. 98.6% for the baseline visit). Thus, compared to graph-based alignment across these samples, up to 8.8% of HBV sequencing data can be missed due to the use of a linear reference sequence of the *correct* HBV genotype compared to the HBV reference graph.

HBV-derived sequences that failed to align to the non-subgenotype B2 reference were also not uniformly distributed across the genome (**S5 Fig in S1 File**). The distribution of where rescued reads aligned to the B2 reference is informative as more reads from non-HBV genotype B references were rescued across the HBV genome except in the pre-core/core region. At this locus, the average distribution of rescued reads from genotype C was always the lowest (**S5 Fig in S1 File**), which is unsurprising as the pre-core/core region within HBV subgenotype B2 reflects a known recombination event between genotypes B and C [43]. For reads which still failed to align to the best performing linear B2 reference sequence across each sample, 30.5%-63.2% were rescued via graph-based alignment (N = 130,154, 71,526, 83,348, 61,692, 52,071 reads rescued, respectively). The distribution in the start sites of all rescued reads was similarly non-uniformly distributed. Interestingly, the loci in which graph-based alignment rescued the most reads also corresponded to loci with increased pairwise nucleotide diversity estimated across the 44 phylogenetically representative proposed HBV reference sequences (**S6 Fig in S1 File**), consistent with our simulations and suggesting regions of increased genetic diversity globally may correspond to loci of increased intra-host sequence variation in real CHB samples.

There was no significant difference in the observed proportion of successfully aligned reads when using all or a down-sampled subset (to 20,000X coverage) of the QC-passed HBV sequencing data across the real CHB samples (P>0.99). Additionally, there was no significant difference in the proportion of aligned reads when the full-length linear reference sequences or extended linear reference sequences were used across the linear reference-based alignments (*p* = 0.76) (**S7 Fig in S1 File**).

### Graph-derived consensus sequences are more genetically similar to HBV sequencing datasets

Unlike the genotype or subgenotype-specific genomes included within an established set of reference sequences, sequencing data-derived consensus sequences should reflect the most accurate representation of the HBV genome causing the infection. While no single full genome-length HBV sequence could realistically capture the sequence variation observed across our simulated high throughput HBV sequencing data, the graph-based variant calling performed using the variation graph toolkit (VG) provided a consensus sequence with the lowest genetic distance to the full set of HBV genomes used in the simulations (**S8 Fig in S1 File**) mash distances ranging between $7.50 \times 10^{-2}$ and $7.82 \times 10^{-2}$. Using the Mash distance as an approximation of average nucleotide identity (ANI), consensus sequences had ANIs ranging between 92.2%-92.5%, with the consensus inferred from VG-based variant calling having the highest ANI (92.5%).

For consensus sequences derived using the subset of HBV sequencing data generated from HBV genotypes B/C only, VG-based variant calling also resulted in the sequence with the lowest Mash distance and highest ANI compared to the full HBV genotype B/C sequences. However, all genotype B and C specific consensus sequences had similar Mash distances, ranging between $6.01 \times 10^{-2}$ and $6.17 \times 10^{-2}$, and ANIs ranged between 93.8% and 94.0% (**S9 Fig in S1 File**).

For analyses of the real CHB sequencing data, the *de novo* assembled viral haplotypes always had the lowest Mash distance compared to the HBV sequencing data for each sample. This

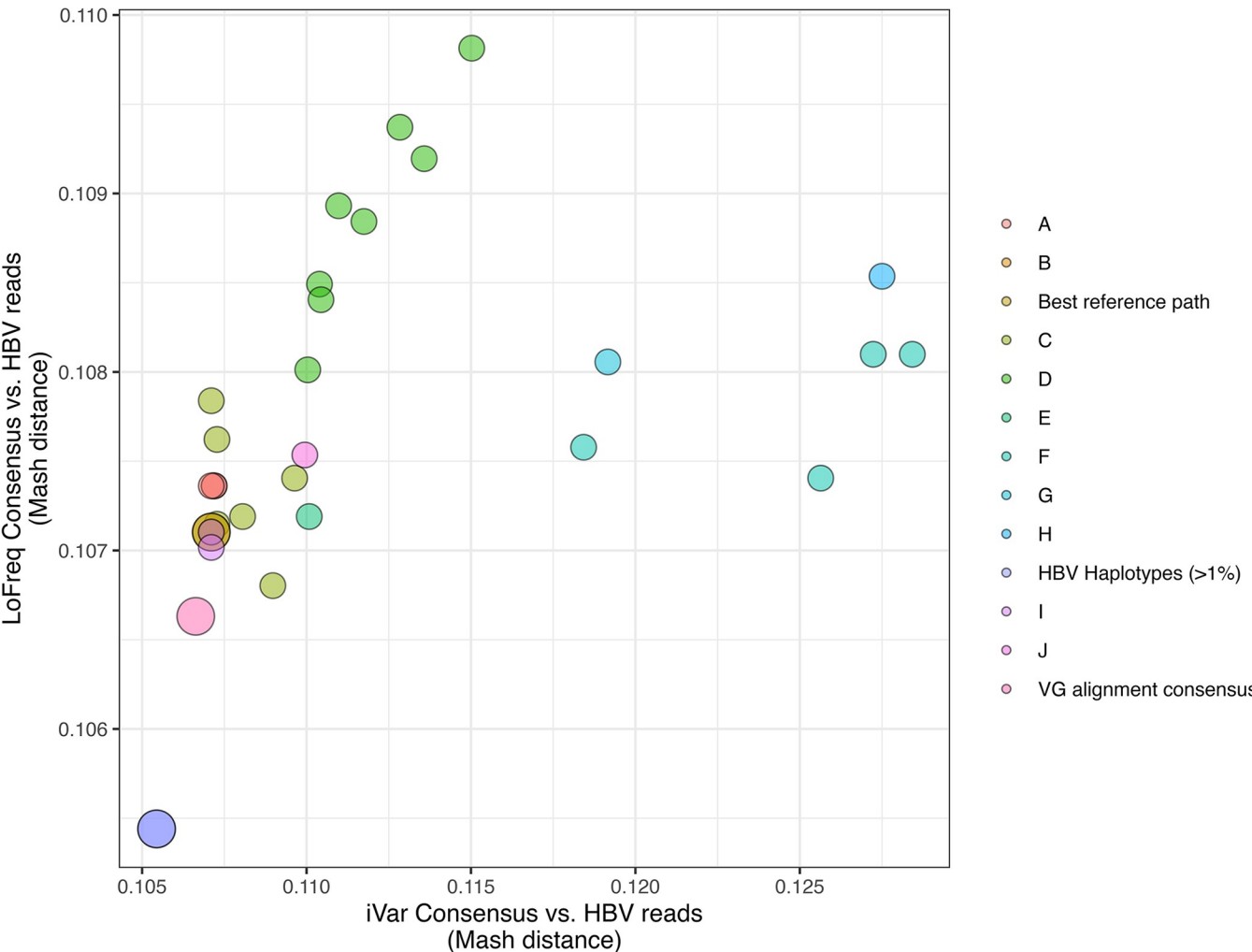

**Fig 3. Genetic distance comparisons of consensus sequences and *de novo* assembled HBV haplotypes with CHB sequencing data from the first CHB sample.** Points reflect the Mash distance estimated between each consensus sequence generated from the 44 HBV reference sequences or the HBV reference graph. The Y axis reflects the Mash distance estimated between each LoFreq-derived consensus sequence and the X axis reflects the Mash distance estimated between each iVar-derived consensus sequence. The color of each point reflects the genotype of the reference used to generate a consensus, or if the consensus was derived via graph-based alignment or reflects sample-specific HBV haplotypes. Points for graph-derived consensus sequences, including VG-based variant calling ('VG alignment consensus'), graph-based surjection ('Best reference path'), and the *de novo* assembled viral strains ('HBV Haplotypes (>1%)') are enlarged.

suggests viral haplotypes comprising an individual's CHB quasispecies better approximate the overall sequence diversity of an infection than any derived consensus sequence.

For consensus sequence comparisons, a graph-based variant calling approach resulted in consensus sequences with the lowest average Mash distance and highest ANI compared to each sample-specific set of HBV sequencing data across the longitudinal CHB samples. Our graph-based consensus sequence construction method provided improvements (i.e., a reduction in genetic distance) over attempts involving linear reference sequences when variants were identified via LoFreq and every sample other than SRR7471499 when consensus sequences were generated using iVar (**Fig 3**, **S10 Fig in S1 File**). For this sample, graph-based variant calling and consensus sequence generation resulted in the same genetic similarity estimate (ANI = 89.3%) as a consensus derived using a subgenotype C1 reference sequence (GenBank ID: DQ089781). While both iVar and LoFreq can be used to identify variants across a

diverse set of viral pathogens [44, 45], LoFreq has repeatedly been used to identify variants from real CHB-derived HBV sequencing data [46, 47]. Additionally, while both consensus identification methods used the same site-specific depth threshold, our LoFreq-based approach accounted for insertions and deletions, potentially explaining its consistently lower Mash distance compared to the consensus sequences obtained via iVar (**Fig 3**, **S10 Fig in S1 File**).

For linear reference iVar-based variant calling across all samples, Mash distances ranged between $1.07 \times 10^{-1}$ and $1.27 \times 10^{-1}$, for SRR7471499, SRR7471500, and SRR7471502. For SRR7471501, distances ranged between $1.07 \times 10^{-1}$ and $1.30 \times 10^{-1}$ and for SRR7471513 ranged between $1.07 \times 10^{-1}$ and $1.28 \times 10^{-1}$. Mash distances from the LoFreq-derived consensus sequences ranged between $1.07 \times 10^{-1}$ and $1.10 \times 10^{-1}$ for SRR7471499, SRR7471501, SRR7471502, and SRR7471513 and $1.07 \times 10^{-1}$ to $1.09 \times 10^{-1}$ for SRR7471500. VG-based variant calling derived consensus sequences each had Mash distances of $1.07 \times 10^{-1}$. Additionally, we observed no differences in the Mash distances between consensus sequences derived using reads re-aligned to a single path within the graph, or surjected, into the HBV subgenotype B2 path (B2, GenBank ID: GU815637) and the linear reference-based alignment derived consensus using the same B2 reference.

## Discussion

In this study, we confirm that the choice of reference plays a critical role in the alignment of high throughput HBV sequencing data and can influence the construction of sample-specific consensus sequences in genetic studies of CHB. We also demonstrate that sequence variation graphs can improve upon widely accepted methodologies used for sequence alignment of HBV. Using both real-world CHB and simulated high-diversity HBV sequencing datasets, we show that alignment to a phylogenetically representative reference graph results in a higher proportion of successful sequence alignment and facilitates the generation of accurate sample-specific consensus sequences.

As the benefits of sequence-to-graph alignment are greatest for highly diverse sequencing datasets, the utility of graph-based sequence alignment is dependent upon the research question of interest. For example, sequence-to-graph alignment recovers only marginally more simulated sequencing data generated from a subset of HBV genotype B and C sequences (**S3 Fig in S1 File**) compared to linear reference-based approaches using genotype B or C reference sequences. Furthermore, linear reference-based sequence alignment is highly successful at capturing HBV sequences from regions across the HBV genome with non-extreme global sequence diversity (**Fig 1**). While our results demonstrate that for regions of increased diversity, any single linear reference is likely insufficient to capture the genetic variation observed across all HBV genotypes/subgenotypes or mixed CHB infections, many CHB infections are comprised of a single HBV genotype, and thus linear reference-based alignment using a correct genotype/subgenotype sequence would not be expected to omit important information. This is, however, not guaranteed, as we find that >8% of viral sequences can fail to align to a phylogenetically representative linear reference sequence and may represent variation within highly diverse regions encoding immunodominant epitopes of biological or clinical importance. For more genetically diverse infections, a hybrid approach in which a linear reference-based alignment is followed by graph alignment of unmapped reads could also solve issues related to reference ambiguity while limiting the computational burden associated with graph-based sequence alignment. Notably, we find reads rescued via graph alignment largely originate from regions across the HBV genome of increased global sequence diversity (**S6 Fig in S1 File**), suggesting these loci could also correspond to regions of increased intra-host genetic variation.

The generation of consensus sequences is an important product of microbial-focused genomic analyses, and novel software and workflows devoted to generating pathogen consensus sequences, including for SARS-CoV-2 [29], continue to be developed. In addition to providing an accurate characterization of the genome comprising a clinical infection, publicly-available consensus sequences enable molecular epidemiology-focused research, allow for large-scale phylogenetic analysis, and can aid disease surveillance efforts [48–51]. Consensus sequences can also serve as the 'reference' sequence in subsequent bioinformatic analyses, reducing the number of spuriously identified variants for HBV [34]. Thus, care should be taken to ensure the most accurate and genetically representative sequences are obtained from clinical CHB or other HBV-infected samples. We show that graph-based alignment and variant calling can often improve upon linear reference-based approaches to derive sample-specific consensus sequences, even when such efforts utilize reference sequences of the correct HBV subgenotype, which produces consensus sequences less genetically similar to an individual's CHB quasispecies than sequences derived via graph alignment. However, given the minute differences in average nucleotide identity between consensus sequences obtained from any of the best-performing linear references and our graph-based approach, the deleterious consequences of linear reference-based alignment are likely minimal when an effort is made to first identify a genetically representative reference sequence for use in alignment.

However, when reference selection is not carefully considered, unrepresentative reference sequences can impact the fidelity of consensus sequences and other downstream phylogenetic-focused analyses [35, 37]. It is, therefore, possible that some publicly available full-length HBV genome sequences, including the 44 reference sequences we used to construct our reference graph, are not the most accurate HBV-related sample-specific consensus sequences. While alternative approaches to infer sample-specific consensus sequences exist [5, 34], our approach using the Mash distance to compare and identify the consensus sequence that best approximates the set of HBV sequencing data or *de novo* assembled haplotypes could provide a mechanism by which sample-specific consensus sequences are compared and selected for use as ideal reference sequences.

Sanger sequencing, alternatively, could be useful if consensus sequences were the sole study objective, as Sanger sequencing largely results in a consensus sequence of amplified primer-guided fragments of HBV sequence. However, this approach lacks the depth required to detect low-frequency variants and the full spectrum of viral diversity present within a host. Additionally, as the primer sequences used to amplify viral DNA are derived from some reference sequence selected *a priori*, the issue of reference bias remains. For example, while primers are typically derived from highly conserved loci, sequence heterogeneity due to mixed/recombinant infection or due to infection with a non-endemic HBV genotype could result in primers being constructed within inappropriate loci and lead to the failed amplification and subsequent lack of coverage across large sections of the HBV genome, biasing any subsequent analysis [32].

While sequence-to-graph alignment requires more computational resources than linear alignment-based approaches, especially for the VG map mapper (**S2 Table in S1 File**), if the goal is to capture and retain as much HBV-related sequencing data as possible for analysis, we show that graph-based methods outperform traditional linear reference-based alignment for HBV. We should note that effective tools enabling sequence-to-graph alignment and the subsequent identification of graph-derived genetic variation are a relatively recent development. Improvements in computational performance have already been demonstrated through graph simplification and the development of more advanced mapping and variant identification models [52–54], with further improvements expected [55].

While alternatives to graph-based alignment, which leverage multiple reference sequences, have also been developed, such as alignment using multiple linear references in tandem [56, 57],

their performance has not been assessed using HBV sequencing data. Furthermore, an added benefit of graph-based approaches is that differences observed between the embedded paths/reference sequences can be utilized during variant calling to identify loci of genetic variation, in addition to mutations inferred from the alignment of sequencing data directly. The ability to leverage graph topology was demonstrated in our use of path depth to infer the genotype of HBV sequences aligned to the reference graph. Alternative graph-based prediction methods, including models for HBV subgenotype prediction or recombination detection, are worth further exploration. Whether metrics linked to graph topology or complexity, including path depth, can be used to better characterize the viral genetics of CHB quasispecies, either within specific regions or across the genome [58, 59], or the genetic diversity of HBV generally remains unexplored. For example, we observe the distribution of path depth within the phylogenetically representative HBV reference graph approximates a universal gene frequency distribution typical of many bacterial species (**S2 Fig in S1 File**) [36], despite there being no distinction between a core and accessory genome for HBV. Future efforts should investigate the utility and potential clinical importance of these graph-derived measures of genetic complexity for HBV and other microbial pathogens of public health importance. For example, graph-to-graph comparisons could enable the analysis of genetic sequence data in ways that Euclidean data structures cannot.

Finally, while sequence variation graphs are a relatively recent advancement, they are increasingly being used to investigate highly genetically diverse microbial pathogens and regions of the human genome, further catalyzed by the recent completion of the first human pangenome reference [60]. In this study, we demonstrate the limitations of using linear HBV reference sequences to derive consensus sequences for CHB samples. Furthermore, we hope to mitigate issues of HBV reference ambiguity by making this HBV reference graph publicly available, which will also promote the use of graph-based advances in genetic analyses to improve our understanding of CHB genetics.

## Materials and methods

### Source of genetic sequence data

**Full-length HBV genome sequences.**  A set of non-redundant full-length HBV genomes (N = 2,837) was obtained from the publicly-available resource provided by McNaughton *et al.* [38]. Briefly, 7,108 full-length HBV genomes were obtained from the HBVdb database (https://hbvdb.lyon.inserm.fr/HBVdb/) and recombinant or highly similar full-length HBV genome sequences were removed. A set of 44 sequences representative of all phylogenetically-identified genotypes, subgenotypes, and genetically-distinct clades was then identified for use as reference sequences in downstream analyses.

**High-throughput CHB sequencing data.**  HBV-targeted sequencing data from an individual included in a longitudinal cohort study of treatment-naïve individuals with CHB was obtained via the NCBI Sequence Read Archive (BioProject ID: 479693) [61]. Sample-level clinical and demographic data were obtained through communication with study authors. The high-throughput CHB sequencing data included in this study reflects five longitudinally sampled visits between 1991–1996 from a single HBeAg positive individual (identifier 'C4'). Sequencing was performed using an Illumina HiSeq 2500, as previously described [61]. Cutadapt was used to trim adapters, poor-quality bases, and reads <36bp long [62]. FastQC was used to ensure the post-QC data passed Illumina sequencing-related QC checks [63].

Patients with HBV infection, including individual C4, were recruited with fully informed written consent from the Division of Gastroenterology and Hepatology at the National University Health System, Singapore, and the original study received approval from local institutional review boards [61].

**Simulated high-throughput sequencing data.** Realistic high throughput HBV sequencing data were simulated using InSilicoSeq, which enables the generation of error-prone Illumina-like sequencing data with pre-specified abundance/coverages [64]. Two datasets of paired-end sequences/reads were generated using an Illumina HiSeq error model, the first set (N = 50,000 reads/genome) was simulated from each of the recommended HBV reference sequences (N = 44). The second set (N = 500,002 reads total) was simulated from a randomly selected HBV genome sequence from each of the 9 HBV genotypes (excluding genotype J, as only a single isolate remains available) and 50 additional randomly selected HBV genomes not included within the HBV reference graph (total N = 59).

## Sequence-to-graph alignment

**HBV reference graph construction and alignment.** A sequence variation graph, termed the HBV reference graph, was created using the full set of phylogenetically representative reference sequences (N = 44) (**S1 File**) [38]. The HBV reference graph was created using the pangenome graph builder (PGGB) pipeline, which performs pairwise whole-genome alignment using wfmash and graph induction using the seqwish software and then sorts and orders the graph via partial order alignment using smoothxg [65–67].

The variation graph toolkit (VG, v1.39) was used to perform all graph-related format conversions, indexing, sequence-to-graph alignment, and collating of mapping/alignment statistics, as described in the VG documentation [33, 68]. Fast short-read alignment via the VG giraffe mapper was accomplished by creating a haplotype-aware graph index where each reference genome was indexed as a unique haplotype [69]. Highly accurate but more computationally intensive graph-based mapping was performed using the VG map mapper.

**Establishing internal validity of the HBV reference graph.** All reads simulated from the graph-embedded HBV genomes were concatenated and then randomly subsampled to 20,000X coverage seven times. Coverage-based subsampling was performed using rasusa [70]. These subsampled HBV sequencing datasets were aligned to the HBV reference graph using the haplotype-aware VG giraffe mapper. For the graph to be internally valid, we required >99% of the reads simulated from HBV genomes embedded within the graph to align successfully.

To assess whether each path within the graph was utilized during sequence alignment and to test whether aligned sequences had the highest alignment scores to graph-embedded HBV genomes, which were more genetically similar to the aligned sequences, each full-length HBV sequence (N = 2,837) was aligned to the graph using VG map. A 'correct' alignment was observed if the reference path with the highest alignment score was of the same HBV genotype as the query sequence. Path-specific alignment scores were also derived for alignments made using the set of simulated high throughput sequencing data from HBV genomes not used in graph construction (N = 59). Briefly, by identifying the graph nodes for each path with successful alignments, the genome path with the most alignments was able to be identified (**S1 Fig in S1 File**). Path-specific alignment scores were derived using the sum of weights estimated for each node involved in a successful alignment. Weights reflect the path depth of each node (i.e., the number of genome sequences containing/traversing through the node), with nodes traversed by a single HBV genome weighted heavily and nodes traversed by all genomes weighted least (**S2 Fig in S1 File**). A 'correct' alignment was observed if the path with the highest weighted alignment score was of the same HBV genotype as the genome sequence used to simulate the HBV sequencing data.

### Alignment of HBV sequencing datasets–graph vs. linear references

**Simulated HBV sequencing data.** To determine whether a graph-based reference improves sequence alignment compared to linear reference-based approaches for HBV sequencing datasets, we aligned the combined simulated high-throughput sequencing data (generated from 59 HBV genomes not included within the graph) to the graph using VG giraffe and to each linear reference sequence (N = 44) using BWA-MEM. The proportion of successfully aligned reads was obtained using 'VG stats' and 'SAMtools flagstat,' [71] respectively. While comparisons of the computational time and resources required for variation graph and linear-reference-based aligners have been performed previously [68], '/usr/bin/time' estimates for the alignments using BWA-MEM, VG giraffe, VG giraffe in fast-mode, and VG map can be found in **S1 Table in S1 File**. To approximate a more realistic scenario in which the observed genetic diversity spans a subset of HBV genotypes known to circulate within a geographic region rather than all currently known HBV genotypes/subgenotypes, linear reference and graph-based alignment comparisons were performed using simulated sequencing data from randomly selected HBV genotype B (N = 6) and C (N = 12) sequences, the primary genotypes endemic in East and Southeast Asia [42, 43].

We estimated the depth of coverage across the HBV genome for the alignment of all simulated high-throughput sequencing data to each linear reference using 'SAMtools depth'. Genotype-specific depth estimates were obtained by estimating the mean alignment depth across alignments made using references of the same genotype via a sliding-window approach (50bp wide) in R. Local minima in depth were estimated using the ggmisc package in R. To approximate site-specific depth of coverage across the HBV genome from the graph-based alignment, the start site of each successfully aligned read was used to infer coverage by estimating a rolling sum of the median number of reads within a sliding window the length of the simulated reads (125bp).

To facilitate the comparison of whether regions of poor coverage corresponded to loci of increased pairwise diversity, the local nucleotide sequence diversity across the set of reference sequences (N = 44) was estimated using a sliding window approach (150bp wide) in R using the pegas package [72].

**Alignment of real CHB sequencing data.** To determine the approximate sequencing depth for each CHB sample (N = 5), raw sequencing data were aligned to each linear reference sequence (N = 44) using BWA-MEM [73]. Alignment quality was assessed using Qualimap (v2.2.1) [74]. The proportion of successfully aligned reads were estimated using 'SAMtools flagstat'. For each sample, the linear reference with the highest proportion of successfully aligned reads was an HBV subgenotype B2 sequence (GenBank ID: GU815637). For alignments to this reference, mean depth of coverage ranged from 82,930X-334,157X.

To reduce computational time and resources required for our analyses, QC-passed reads were down-sampled to obtain an average coverage of 20,000X. To test whether subsampling altered the proportion of successfully aligned reads, subsampled reads were also aligned to each linear reference sequence and the proportion of successful alignments was compared to the alignments involving all QC-passed sequencing data using a binomial generalized linear mixed model (GLMM) with random intercepts in R. The GLMM treated each alignment as a binomial outcome (successful alignment vs. not successful alignment), with the total number of reference-specific alignments used as weights. Whether alignments of these subsampled reads to an extended linear reference, obtained by concatenating the first 120bp of each reference to the end of each sequence, altered alignment statistics were also assessed using the same GLMM performed in R. To identify whether reads which failed to align to sub-optimal linear references (non-HBV subgenotype B2) were uniformly distributed across the genome,

unaligned reads from each linear reference-based alignment were re-aligned to the best performing linear reference. The genome-wide distribution of these 'rescued' reads was visually assessed in R.

Graph-based alignment of subsampled CHB sequencing data was performed using the VG map mapper. For samples with higher alignment proportions to the graph than any linear reference, unmapped reads from the best performing linear reference for each sample were re-mapped to the graph and the genome-wide distribution of the reads 'rescued' via graph alignment was visualized using R. To identify and visualize the loci where HBV sequence was rescued via graph alignment, the rescued reads were queried via BLAST against a compacted de Bruijn Graph comprised of the reference sequences and *de novo* (reference-free) assembled HBV haplotypes from each sample created using Bifrost and visualized with Bandage [75, 76]. We also performed BLAST in Bandage using these successfully re-mapped reads against the HBV reference graph only to confirm that rescued reads mapped to regions of increased graph complexity.

## Derived consensus sequences–graph vs. linear reference sequences

**Simulated HBV sequencing data.** Consensus sequences were obtained from linear reference-based alignments of simulated non-graph derived sequencing data using iVar [77]. iVar was developed to analyze amplicon-based viral sequencing data and leverages SAMtools to call variants and derive a consensus from the most common nucleotide across each position in an alignment file. We used a minimum base-level quality score of 20 and depth threshold of 10 while accounting for ambiguous nucleotides. For graph-based alignments, we used a wholly graph-based variant calling approach leveraging the alignments across all paths using VG [33], followed by consensus generation via bcftools [78].

**Longitudinal CHB sample consensus sequences.** Prior to performing alignment and variant calling for the real CHB samples, QC-passed paired-end reads were merged using PEAR and filtered to retain the highest quality reads >150bp long for analysis via bbmap [79, 80]. Reads were aligned to each linear reference or the HBV reference graph, followed by iVar-based consensus sequence identification. We also performed variant calling using the LoFreq software, a variant calling tool able to identify even low-frequency variants from high-coverage data across diverse genetic sequencing datasets [81], for each linear reference-based alignment followed by consensus generation using a majority allele rule for each site (i.e. alleles with frequency >50% were integrated into the consensus sequence) via bcftools. For LoFreq-derived consensuses, we used the same depth threshold (10) used in iVar and estimated insertion/deletion qualities which were used in addition to LoFreq's method of combining base-level, mapping, and alignment quality information to determine variant quality and identify the majority nucleotide at each position, accounting for insertions/deletions. Graph-based alignment was performed using VG giraffe. VG-based variant calling using the graph alignments and consensus generation were obtained via bcftools. For these CHB samples, we also derived consensus sequences after re-aligning the successful graph-aligned reads to a single path within the graph via VG, termed surjection, followed by iVar consensus construction. Graph-aligned reads were surjected into the path corresponding to the best-performing linear reference.

## Consensus sequence comparisons

**Consensus sequence comparisons from simulated HBV sequencing data.** Comparisons between each simulation-based consensus sequence and the full set of HBV genomes from which reads were simulated were performed using Mash [82], which estimates a genetic distance metric, the Mash distance, based on the estimated mutation rate between two sets of

sequences and the Jaccard index (the fraction of k-mers shared between the comparison sequences). The Mash distance also approximates average nucleotide identity (ANI) estimates, with ANI equivalent to one minus the distance estimate, while also having the benefit of facilitating comparisons between sequences/sequencing datasets of variable lengths/sizes [82]. Given the short length of the HBV genome (3.2kb), a k-mer sequence length of 7 was used for Mash distance estimations [83, 84]. The consensus sequence with the lowest estimated genetic distance with the set of full-length HBV genome sequences can be inferred to be the most accurate or genetically representative consensus sequence.

**Identifying accurate consensus sequences from real CHB sequencing data.** To facilitate comparisons between CHB-derived consensus sequences and to identify the most genetically similar consensus to the HBV quasispecies of each sample, we estimated the Mash distance between each consensus and the subsampled HBV sequencing data which aligned to the best performing reference (linear or graph-based) for each sample. We also performed *de novo* HBV strain-level assembly using SAVAGE and VG-Flow to identify the viral haplotypes comprising each CHB infection [85, 86]. For each sample, the best-performing linear reference was added to the SAVAGE output for VG-Flow to improve strain-level contiguity and assembly. The set of sample-specific viral haplotypes with frequencies >1% were included in all pairwise genetic distance comparisons. The consensus sequence with the lowest estimated genetic distance with the HBV-specific high throughput sequencing data can be inferred to be the most accurate and genetically representative consensus sequence for each sample.

**Statistical analysis.** Read depth and coverage-related statistical estimates were performed using SAMtools while the generalized linear mixed model regressions, sliding-window-based analyses, and visualizations were performed in R(v4.2.2). Unless otherwise specified, default parameters were used for the variant calling, consensus construction, and sequence comparison tools described above. For all analyses, a statistical significance threshold of $p<0.05$ was used.

## Supporting information

**S1 File.**
(PDF)

## Author Contributions

**Conceptualization:** Dylan Duchen, Priya Duggal, Genevieve L. Wojcik.

**Data curation:** Dylan Duchen, Genevieve L. Wojcik.

**Formal analysis:** Dylan Duchen, Priya Duggal, Genevieve L. Wojcik.

**Funding acquisition:** David L. Thomas, Priya Duggal, Genevieve L. Wojcik.

**Investigation:** Dylan Duchen, Priya Duggal, Genevieve L. Wojcik.

**Methodology:** Dylan Duchen, Priya Duggal, Genevieve L. Wojcik.

**Project administration:** Priya Duggal, Genevieve L. Wojcik.

**Resources:** Dylan Duchen, David L. Thomas, Priya Duggal, Genevieve L. Wojcik.

**Software:** Dylan Duchen, Priya Duggal, Genevieve L. Wojcik.

**Supervision:** Candelaria Vergara, Chloe L. Thio, David L. Thomas, Priya Duggal, Genevieve L. Wojcik.

**Validation:** Dylan Duchen, Priya Duggal, Genevieve L. Wojcik.

**Visualization:** Dylan Duchen.

**Writing – original draft:** Dylan Duchen, Priya Duggal, Genevieve L. Wojcik.

**Writing – review & editing:** Dylan Duchen, Steven J. Clipman, Candelaria Vergara, Chloe L. Thio, David L. Thomas, Priya Duggal, Genevieve L. Wojcik.

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
