## [Decision Letter · Decision Letter 0]

3 Jan 2024

PONE-D-23-40273A hepatitis B virus (HBV) sequence variation graph improves alignment and sample-specific consensus sequence constructionPLOS ONE

Dear Dr. Duchen,

Thank you for submitting your manuscript to PLOS ONE. After careful consideration, we feel that it has merit but does not fully meet PLOS ONE’s publication criteria as it currently stands. Therefore, we invite you to submit a revised version of the manuscript that addresses the points raised during the review process in order to ameliorate the manuscript.

We look forward to receiving your revised manuscript.

Kind regards,

Isabelle Chemin, PhD

Academic Editor

PLOS ONE

Journal Requirements:

Reviewers' comments:

Reviewer's Responses to Questions

**Comments to the Author**

1. Is the manuscript technically sound, and do the data support the conclusions?

Reviewer #1: Partly

2. Has the statistical analysis been performed appropriately and rigorously? 

Reviewer #1: Yes

3. Have the authors made all data underlying the findings in their manuscript fully available?

Reviewer #1: Yes

4. Is the manuscript presented in an intelligible fashion and written in standard English?

Reviewer #1: Yes

5. Review Comments to the Author

Reviewer #1: The authors proposed a graph for sequence alignment of sequences generated by NGS, to improve alignment and to produce a better specific consensus sequence. The manuscript is not clearly understandable for a wide public.

1. The main problems exposed by the authors refer to sequenced obtained by NGS, and not by Sanger sequencing, and this is not specified in the text, nor in the Abstract. The authors should discuss if they think that sequences generated by Sanger are subjected to these same limitations?

2. Introduction, page 9, line 12: a mutation rate higher than the one found for most DNA viruses.

3. Line 19. The authors cited a reference of 1997 to argue that the consensus sequence is not suitable to analyze transmission events in HBV. However, in the study cited, they used a sequence of 263 bp to document the transmission event. Other factors, such as using a larger genomic fragment to report the transmission event, more than quasispecies, can be the cause of the confusion reported in the cited article. It might be advisable to eliminate this argument (ref 30).

4. A short explanation of the aim of the study should be included at the beginning of the results.

5. Figure 1B. The authors should include smaller numbers in the Y axis scale of the coverage, to evaluate more properly the drops in coverage.

6. The ¨middle section¨ of Figure 1B (could be named 1C) refers to nucleotide diversity respect to what sequence?

7. Mats and Methods. Lines 332-342. No Ethical approval is provided, only informed consent, but the authors described many details of a patient and performed sequencing on his samples. So the Ethics statement of page 4 is not completely correct.

8. Please revise the accuracy of the references; the number of references is also elevated.

9. Some of the sequences used as references for this study were generated with the same limitations explained in this study. The authors should discuss this.

10. In general, the authors should try to provide some simpler message for the whole audience, both in Abstract, Introduction and Discussion.

6. PLOS authors have the option to publish the peer review history of their article (what does this mean?). If published, this will include your full peer review and any attached files.

Reviewer #1: No

---

## [Author Response · Author response to Decision Letter 0]

6 Mar 2024

Please see our response to reviewers below, we have also uploaded a PDF version of these comments.

Thank you for the thoughtful comments from the reviewer. We appreciate that they recognized the utility of our proposed graph-based approach and believe our updated text is more easily understandable to the broad readership of PLOS ONE.

We have addressed each of the comments in our response below and in the manuscript, as noted. 

Editorial comments

Reviewer #1: The authors proposed a graph for sequence alignment of sequences generated by NGS, to improve alignment and to produce a better specific consensus sequence. The manuscript is not clearly understandable for a wide public.

We appreciate the reviewer’s comments, and thank them for their time and close reading of our manuscript. We agree that our text could have been more approachable for those unfamiliar with these methods. We have revised the text to provide a more detailed introduction to the utility of graph genomes, particularly in the context of microbial- and human-focused genetic analyses. We would like to note that while our work addresses what seems to be a niche bioinformatics issue of reference bias due to a mismatch between specific inputs of bioinformatics software, the problem we solve via the use of sequence variation graphs for HBV mirrors the consensus among genomic researchers that pangenomic graph-based reference sets both improves the accuracy of genetic analyses while addressing a major issue of equity and inclusion which is pervasive across the field of genetics. Considerable effort and investment has been made by the National Human Genetics Research Institute (NHGRI) to support the development and use of graph-based tools similar to those we describe in our work (e.g., The Human Pangenome Reference Consortium project). Therefore, while nascent, we believe that sequence variation graph tools and bioinformatics workflows involving such approaches will play an increasingly important role in the analysis of both microbial and human genetic data. 

1. The main problems exposed by the authors refer to sequences obtained by NGS, and not by Sanger sequencing, and this is not specified in the text, nor in the Abstract. The authors should discuss if they think that sequences generated by Sanger are subjected to these same limitations?

We agree with the reviewer that it is essential to discuss the implications of Sanger sequencing in this context. To address this, we have added a discussion to the manuscript that examines the potential limitations of Sanger sequencing in capturing within-host viral diversity. While it is true that Sanger sequencing can provide a consensus sequence that is representative of the predominant virus in a sample, it lacks the depth required to detect low-frequency variants and the full spectrum of viral diversity present within a host. This is a critical shortfall, especially when exploring the evolutionary dynamics of HBV, wherein the detection of minor variants can have significant implications for disease progression, treatment response, and resistance development.

Furthermore, we also consider the reviewer's concern regarding reference bias in Sanger sequencing. We agree that while Sanger sequencing may yield longer reads that could benefit consensus sequence assembly, the technique is not without its limitations, namely potential biases introduced during primer design and amplification.(PMID: 31068626) We have now included discussion how primer design based on conserved loci can be influenced by HBV genetic heterogeneity and how this could lead to biased results, particularly when dealing with mixed or recombinant infections or non-endemic HBV genotypes.

The revisions underscore the necessity of careful primer design and the importance of acknowledging the potential for reference bias, irrespective of the sequencing method used. Nonetheless, NGS offers unparalleled advantages in terms of depth and resolution for within-host diversity studies, which is the primary rationale behind our study's focus on NGS methods.

Please see our newly included text within the manuscript, found on lines 292-302 (Simple Markup):

“Sanger sequencing, alternatively, could be useful if consensus sequences were the sole study objective, as sanger sequencing largely results in a consensus sequence of amplified primer-guided fragments of HBV sequence. However, this approach lacks the depth required to detect low-frequency variants and the full spectrum of viral diversity present within a host. Additionally, as the primer sequences used to amplify viral DNA are derived from some reference sequence selected a priori, the issue of reference bias remains. For example, while primers are typically derived from highly conserved loci, sequence heterogeneity due to mixed/recombinant infection or due to infection with a non-endemic HBV genotype could result in primers being constructed within inappropriate loci and lead to the failed amplification and subsequent lack of coverage across large sections of the HBV genome, biasing any subsequent analysis.[36]”

2. Introduction, page 9, line 12: a mutation rate higher than the one found for most DNA viruses.

We are unsure what the reviewer means by this comment. If they are questioning the validity of the text, we base this claim on the following reference (PMID: 30268787). 

 “HBV is more diverse than other dsDNA viruses (Figure 3A) with a level of variation and rate of evolution that is more comparable to an RNA virus than a DNA virus”

To avoid potential confusion and clarify the text, we have changed “...most DNA viruses” to “...most double stranded DNA viruses”, now found on lines 12-13.

3. Line 19. The authors cited a reference of 1997 to argue that the consensus sequence is not suitable to analyze transmission events in HBV. However, in the study cited, they used a sequence of 263 bp to document the transmission event. Other factors, such as using a larger genomic fragment to report the transmission event, more than quasispecies, can be the cause of the confusion reported in the cited article. It might be advisable to eliminate this argument (ref 30).

We thank the reviewer for this note. We agree that more recent evidence should be cited, especially considering current viral genomic assemblers and variant calling workflows are capable of inferring and phasing low-frequency variants across viral genomes using high-throughput short-read sequencing data. However, we believe it is important to reiterate that using consensus sequences alone to infer transmission events can be misleading. So as not to so heavily rely on this one example, we have updated our text to be broadly applicable and have updated our relevant citations to include studies finding the transmission of low-frequency viral strains in HBV (PMID: 28483405) and other viral infections like SARS-CoV-2 (PMID: 33688063).

The updated text on lines 18-20 now reads as follows: 

“...and confound molecular epidemiology or surveillance efforts (e.g., transmission of non-consensus/minor-frequency viral strains).[30–32]”

4. A short explanation of the aim of the study should be included at the beginning of the results.

We thank the reviewer for this note, and have included a section of text at the end of the Introduction, immediately prior to Results(lines 75-78), to frame our results within the context of our overall study aims

“The aim of this study is to assess whether a pangenomic graph-based approach can improve sequence alignment and mitigate reference ambiguity for genetic studies of HBV. Additionally, we aim to determine if this approach leads to more accurate sample-specific consensus sequences compared to standard linear reference-based approaches.”

5. Figure 1B. The authors should include smaller numbers in the Y axis scale of the coverage, to evaluate more properly the drops in coverage.

We appreciate the reviewer's suggestion to improve readability/inference of our figure. We have updated the top panel of this figure to include breaks at intervals of 2,500.

6. The ¨middle section¨ of Figure 1B (could be named 1C) refers to nucleotide diversity respect to what sequence?

We apologize to the reviewer for not making this clear in the text or figure legend. We have updated the legend text to indicate that we are plotting the mean nucleotide diversity estimated from every possible pairwise comparison of the 44 genomes in a sliding window across the genome.

7. Mats and Methods. Lines 332-342. No Ethical approval is provided, only informed consent, but the authors described many details of a patient and performed sequencing on his samples. So the Ethics statement of page 4 is not completely correct.

We thank the reviewer for this comment. We have updated the text on lines 362-364 to indicate that the original study for which sequencing was performed did obtain approval from local institutional review boards and informed consent given. We also agree that many of the demographic details we provide for the longitudinally followed individuals are extraneous and irrelevant to our results. We have therefore removed much of this information within the section now on lines 356-357. 

8. Please revise the accuracy of the references; the number of references is also elevated.

We appreciate the reviewer's comment, and apologize for the inappropriate truncation of several references. We have removed several redundant references, but believe the number of citations accurately reflect the source material and wide array of computational software we have used to complete this work.

9. Some of the sequences used as references for this study were generated with the same limitations explained in this study. The authors should discuss this.

This is an excellent point, however as we do not have access to the raw sequencing data for these sequences we are unable to determine whether a graph-based consensus generation approach results in a different consensus sequence for its constitutive sequences. The interrogation of genbank samples which have linked high throughput sequencing data could be re-analyzed to determine the robustness of the inferred sequences. Regardless, as we agree with the reviewer that this oversight is worth addressing within the text, we have revised the text on the following lines 283-286:

“It is, therefore, possible that some publicly available full-length HBV genome sequences, including the 44 reference sequences we used to construct our reference graph, are not the most accurate HBV-related sample-specific consensus sequences.”

10. In general, the authors should try to provide some simpler message for the whole audience, both in Abstract, Introduction and Discussion.

We thank the reviewer for emphasizing the importance of making our manuscript accessible to a broad audience, including those who may not be familiar with the intricate details of graph-based approaches and bioinformatics. We wholeheartedly agree that the value of scientific research is greatly enhanced when communicated effectively to a diverse range of readers. We have attempted to provide a more easily understandable introduction to our graph-based approach and simplified our language in our revised abstract, introduction, and discussion . While sequence variation graphs and related structures are likely unfamiliar to readers outside bioinformatics and related disciplines, these approaches are gaining popularity and widespread acceptance, further catalyzed by the recent completion of the first human pangenome reference (PMID: 37165242). In our updated text, we take care to explain in straightforward terms the importance of this milestone and how graph-based approaches, including our own work, contribute to its utility and understanding.

---

## [Decision Letter · Decision Letter 1]

11 Mar 2024

A hepatitis B virus (HBV) sequence variation graph improves alignment and sample-specific consensus sequence construction

PONE-D-23-40273R1

Dear Dr. Duchen,

We’re pleased to inform you that your manuscript has been judged scientifically suitable for publication and will be formally accepted for publication once it meets all outstanding technical requirements.

Kind regards,

Isabelle Chemin, PhD

Academic Editor

PLOS ONE

Additional Editor Comments (optional):

Reviewers' comments:

Reviewer's Responses to Questions

**Comments to the Author**

1. If the authors have adequately addressed your comments raised in a previous round of review and you feel that this manuscript is now acceptable for publication, you may indicate that here to bypass the “Comments to the Author” section, enter your conflict of interest statement in the “Confidential to Editor” section, and submit your "Accept" recommendation.

Reviewer #1: All comments have been addressed

2. Is the manuscript technically sound, and do the data support the conclusions?

Reviewer #1: Yes

3. Has the statistical analysis been performed appropriately and rigorously? 

Reviewer #1: Yes

4. Have the authors made all data underlying the findings in their manuscript fully available?

Reviewer #1: Yes

5. Is the manuscript presented in an intelligible fashion and written in standard English?

Reviewer #1: Yes

6. Review Comments to the Author

Reviewer #1: The authors have addressed satisfactorely the comments of the reviewers. All the concerns were responded.

7. PLOS authors have the option to publish the peer review history of their article (what does this mean?). If published, this will include your full peer review and any attached files.

Reviewer #1: No
